# Evaluation of Different Contrast Agents for Regional Lung Perfusion Measurement Using Electrical Impedance Tomography: An Experimental Pilot Study

**DOI:** 10.3390/jcm12082751

**Published:** 2023-04-07

**Authors:** Thomas Muders, Benjamin Hentze, Steffen Leonhardt, Christian Putensen

**Affiliations:** 1Department of Anesthesiology and Intensive Care Medicine, University Hospital Bonn, 53127 Bonn, Germany; mail@benjaminhentze.de (B.H.); christian.putensen@ukbonn.de (C.P.); 2Chair for Medical Information Technology, RWTH Aachen University, 52074 Aachen, Germany; leonhardt@hia.rwth-aachen.de

**Keywords:** individualized therapy, electrical impedance tomography, bedside monitoring, bedside visualization, pulmonary perfusion, contrast agent, functional imaging

## Abstract

Monitoring regional blood flow distribution in the lungs appears to be useful for individually optimizing ventilation therapy. Electrical impedance tomography (EIT) can be used at the bedside for indicator-based regional lung perfusion measurement. Hypertonic saline is widely used as a contrast agent but could be problematic for clinical use due to potential side effects. In five ventilated healthy pigs, we investigated the suitability of five different injectable and clinically approved solutions as contrast agents for EIT-based lung perfusion measurement. Signal extraction success rate, signal strength, and image quality were analyzed after repeated 10 mL bolus injections during temporary apnea. The best results were obtained using NaCl 5.85% and sodium-bicarbonate 8.4% with optimal success rates (100%, each), the highest signal strengths (100 ± 25% and 64 ± 17%), and image qualities (r = 0.98 ± 0.02 and 0.95 ± 0.07). Iomeprol 400 mg/mL (non-ionic iodinated X-ray contrast medium) and Glucose 5% (non-ionic glucose solution) resulted in mostly well usable signals with above average success rates (87% and 89%), acceptable signal strength (32 ± 8% and 16 + 3%), and sufficient image qualities (r = 0.80 ± 0.19 and 0.72 ± 0.21). Isotonic balanced crystalloid solution failed due to a poor success rate (42%), low signal strength (10 ± 4%), and image quality (r = 0.43 ± 0.28). While Iomeprol might enable simultaneous EIT and X-ray measurements, glucose might help to avoid sodium and chloride overload. Further research should address optimal doses to balance reliability and potential side effects.

## 1. Introduction

Efficient gas exchange results from the close matching of regional pulmonary ventilation and perfusion [1]. Ventilation/perfusion matching in the lungs varies physiologically dependent on body position due to changes in gravity influence [1]. It may also be impaired in critically ill patients [1,2]. To re-normalize regional ventilation and perfusion distribution and thus improve gas exchange, mechanical ventilation and positioning therapy may become necessary [2]. Monitoring the regional ventilation and perfusion distribution therefore might be beneficial to individually optimize therapy.

In current clinical practice, global measurements such as blood gas analysis and capnography are used as surrogates for ventilation/perfusion matching [3]. Regional information can currently only be obtained by radiological or nuclear imaging techniques, such as computed tomography (CT) [4] and single-photon (SPECT) [5] or positron emission tomography (PET) [4]. However, these methods are time-consuming, involve transportation risks and radiation exposure, and are not available at the bedside.

Electrical impedance tomography (EIT) is a non-invasive, radiation-free technology that can be used at the bedside. It is well validated for monitoring regional ventilation distribution [6,7]. Meanwhile, several approaches have been proposed to also measure regional perfusion distribution in the lungs [6,7].

The perfusion-related EIT signal results from transient changes in blood conductivity after indicator injection [6]. First-pass kinetics of the indicator can be analyzed to extract regional lung perfusion images using different algorithms [6]. To date, sodium chloride solutions have been used as contrast agents. In 1992, Brown et al. used an isotonic saline (NaCl 0.9%) solution as an indicator [8]. Hypertonic (1 mmol/mL) sodium solution (NaCl 5.85%) was applied by Frerichs et al. [9]. Since then, hypertonic saline has been successfully used as standard for EIT-based lung perfusion measurement in experimental [10,11,12,13,14] and clinical [15,16,17] studies. However, the benefits of hypertonic saline must be weighed against possible risks and side effects during bolus injection. Whereas sodium load might affect vascular resistance and myocardial contractility [18], chloride overload and hyperchloremia are suspected of causing organ dysfunction such as acute kidney injury (AKI) due to impaired renal perfusion [19]. These aspects warrant the investigation of clinically approved fluids to determine their suitability as contrast agents for EIT-based regional lung perfusion measurement. Such studies have so far only been carried out in-vitro by Hellige et al. [20]. 

In this experimental pilot study, we investigated the in-vivo suitability of different injectable solutions as contrast agents for indicator-based regional lung perfusion measurement using EIT.

## 2. Animals and Methods

### 2.1. Animal Preparation and Ethics

After approval from the local ethical committee five healthy (2 female, 3 male; 41 ± 5 kg; 13 ± 1 weeks old) pigs were anesthetized and instrumented as previously described [5,21]. Mechanical ventilation (Evita XL, Draeger Medical GmbH, Luebeck, Germany) was applied in a supine position using a volume-controlled mode at a PEEP = 5 cm H_2_O. Tidal volume was 6–8 mL/kg of body weight. Minute ventilation was adjusted to achieve normocapnia. A fraction of inspired oxygen (FiO_2_) of 0.5 was used.

The pigs were purchased from the local research farm of the University of Bonn, Germany, and were derived from several long-standing colonies (mixed breeds). Animal housing, care, and experiments were performed in adherence with the Guide for the Care and Use of Laboratory Animals [22]. Animals had free access to food and water until 12 h before the beginning of experiments. Transportation to the lab was performed immediately before the beginning of the experiments. Well established medication schemes [5,21] were used for analgosedation and euthanasia. We used EIT as a non-invasive imaging technique for functional lung imaging (refinement) and a repeated-measurement design comparing different contrast agents in a randomized order to reduce the number of animals (reduction). Replacement was not possible in this experimental in-vivo study.

Animal Research Reporting of In Vivo Experiments (ARRIVE) guidelines (Appendix A) were followed.

### 2.2. Measurement and Data Acquisition

#### 2.2.1. Electrical Impedance Tomography

EIT (PulmoVista 500, Draeger Medical GmbH, Luebeck, Germany) was conducted using a 16 electrode belt in adjacent drive configuration at 125 kHz and a frame rate of 50 Hz [11]. To access regional lung perfusion, a 10 mL bolus of the respective contrast agent was injected into a central venous line under temporary apnea (60 s) [11] at PEEP of the preceding ventilation. EIT measurements were stored for further offline analyses.

#### 2.2.2. Contrast Agents

The following five fluids were used as contrast agents:NaCl 5.85%; ready-to-use hypertonic (1 mmol/mL) sodium-chloride solution, clinically used for the substitution of sodium (Natriumchlorid 5.85% Braun Mpc, B. Braun Melsungen AG, Melsungen, Germany).NaBic (sodium-bicarbonate) 8.4%, ready-to-use hypertonic buffer solution, clinically used for correction of the acid-base balance (Natriumhydrogencarbonat 8.4%, B. Braun Melsungen AG, Melsungen, Germany).Jonosteril; ready-to-use iso-tonic, balanced crystalloid solution, clinically used for intravascular volume replacement (Jonosteril^®^, Fresenius Kabi, Bad Homburg, Germany).Glucose 5%; ready-to-use ion-free isotonic glucose solution, clinically used as carrier solution for intravenous drug administration (Glucose 5%, B. Braun Melsungen AG, Melsungen, Germany).Iomeprol 400 mg/mL; ready-to-use non-ionic but iodinated X-ray contrast medium, clinically used for angiography (Imeron^®^ 400 MCT, Bracco Imaging Deutschland GmbH, Konstanz, Germany).

#### 2.2.3. Experimental Protocol

In each pig, three sections of measurements were conducted. Each section consisted of five blocks (one block for each contrast agent) including three repeated bolus injections. The five contrast agents were tested in randomized order (Figure 1). Block randomization was performed separately for any section using sealed envelopes.

### 2.3. Data Processing and Analyses

#### 2.3.1. Extraction of Regional Perfusion Images

EIT data was processed using a custom-made software (MATLAB 2021a, The MathWorks Inc., Natick, MA, USA) and the EIDORS toolbox [23]. Image reconstruction applying a Gauss–Newton algorithm [23] and low-pass filtering to suppress cardiac activity were performed as previously described [11]. Apnea periods were selected for bolus signal analyses (Figure 2a) and apnea drift was corrected using pixel-wise subtraction of an extrapolated linear fit obtained before injection [10,11]. EIT perfusion images were extracted using the Gamma Decomposition (GD) algorithm as recently reported in detail [11]. Briefly, a gamma-variate model was fitted to any regional conductivity–time curve. The source signals of the right heart, the lungs, and the left heart were separated by clustering. After applying these source signals to the regional conductivity–time curves, regional lung perfusion signals can by isolated. Images of relative lung perfusion distribution can then be reconstructed from the lung perfusion signals using the maximum-slope technique [11]. 

#### 2.3.2. Success Rate

In order to extract perfusion images using the GD algorithm, sufficient signal quality as well as a high signal-to-noise ratio are required. Insufficient raw data quality causes the calculation to be aborted, e.g., due to the instability of the fitting or the clustering algorithm. To assess the technical suitability of each contrast agent, the success rate of data extraction was calculated and expressed as a percentage of injections for each animal.

#### 2.3.3. Regional Signal Strength

In order to obtain a measure of signal strength, the maximum amplitude of the regional lung perfusion image was calculated and normalized. For normalization, the average maximum amplitude of all NaCl 5.85% perfusion images was set to 100%. For all contrast agents, the normalized maximum amplitude was quantified regardless of its direction. The normalized amplitude was defined as signal strength.

#### 2.3.4. Image Quality

NaCl 5.85% served as a reference. Comparisons were made separately for each section. Within each section, a mean image was calculated from the three images of the NaCl 5.85% block. For each section, the quality of the images was evaluated by computing the Pearson correlation coefficient between the mean image and any other EIT perfusion image on a pixel-by-pixel basis. For linear correlation, a common mask was derived by logical disjunction (OR) of the individual masks of both images.

### 2.4. Statistical Analyses

For this exploratory study, no reliable pilot data or data from publications were available. Thus, a sample size calculation was not possible. Results are presented using descriptive statistics. Success rate and signal strength are given in percentages. Signal quality is expressed by linear correlation coefficients (Pearson). Data are given as scatter plots and mean and standard deviation, as indicated. Linear correlation (Pearson) was used to compare image quality. A mixed-effects analysis (mixed models for repeated measurements) comparing signal strength between different contrast agents and between different time points was used to show reproducibility of measurements over time. *p* < 0.05 was assumed to be statistically significant. MATLAB (MATLAB 2021a, The MathWorks Inc., Natick, MA, USA) and GraphPad Prism (Version 9; GraphPad Software, San Diego, CA, USA) were used for calculations.

## 3. Results

### 3.1. Success Rate of Perfusion Image Extraction

All animals completed the entire study protocol, which lead to a total of 225 bolus injections being recorded. Video tracings showing the isolated lung-dependent perfusion signals as extracted by the GD algorithm are available from the online supplement for all 225 injections (Appendix A).

All bolus injections of NaCl 5.85% and NaBic 8.4% were suitable for proper perfusion image extraction in each pig (Table 1). Glucose 5% worked well in four of five animals, but the success rate was reduced to 44% in a single pig (Animal 3; only four of nine injections could be analyzed). Comparably, reliable evaluations were feasible after most injections of Iomeprol. Here, the success rate was slightly reduced in one animal (Animal 1) and poor in another (Animal 3). In contrast, signal extraction failed in more than half of all Jonosteril injections.

### 3.2. Global Conductivity Changes and Regional Signal Strength

Global changes in conductivity after bolus injection were different for all contrast agents used (Figure 2a). The highest amplitude was observed for NaCl 5.85% followed by NaBic 8.4%. Jonosteril caused the smallest global signal (Figure 2a). Global amplitudes of Glucose 5% and Iomeprol were in between (Figure 2a) but negative. Consequently, the strength of the extracted perfusion-related regional signal was dependent on the contrast agents used (Figure 2b). The strongest signals were observed for NaCl 5.85% followed by NaBic 8.4%. Jonosteril caused the weakest signal (Figure 2b). Signal strengths of Glucose 5% and Iomeprol were in between (Figure 2b).

For each contrast agent, results on signal strength were reproducible over the entire experimental period. A mixed-effect analysis (mixed models for repeated measurements) showed that the differences in signal strength depended on the contrast agent, but not on time (factor contrast agent: *p* < 0.001; factor time: *p* = 0.28; interaction contrast agent*time: *p* = 0.76, Appendix A).

### 3.3. Regional Perfusion Images

Figure 3 shows representative EIT-based lung perfusion images reconstructed from bolus injections of the different contrast agents for all five animals. Injection of NaCl 5.85% and NaBic 8.4% showed clear and similar results. Perfusion-related signals could be detected and separated for both lungs in all animals. Deviating results can be observed after bolus injection of Glucose 5% and Iomeprol in individual animals. In some cases, there are obvious signal differences between the two sides. For Jonosteril, serious divergences were observed in most cases. In some cases, the perfusion-dependent signals of one side were almost completely absent.

### 3.4. Quality of Regional Perfusion Images

Similarity of the regional lung perfusion images with a reference image was used as a surrogate for image quality. Image quality was excellent for all bolus injections of NaCl 5.85% and NaBic 8.4% in all animals (Figure 4a). For Glucose 5% and Iomeprol image quality was good in most but limited in single pigs (Figure 4a). All images obtained from Jonosteril injections showed limited or poor quality (Figure 4a). Image quality showed a wide scatter when signal strength was low (Figure 4b). As the signal strength increased, the image quality improved in a non-linear manner (Figure 4b). With higher signal strength, image quality was always high, and scatter was minimal. Thus, for NaCl 5.85% and NaBic 8.4% injections, image quality was nearly independent from signal strength (Figure 4b).

## 4. Discussion

In this experimental pilot study, we investigated the suitability of different injectable solutions as contrast agents for indicator-based regional lung perfusion measurement using EIT. The used indicators resulted in both positive and negative changes in global conductivity with significant differences in regional signal strength and image quality.

We recall that the perfusion-related signal results from transient changes in blood conductivity after indicator injection [6]. Depending on the electrochemical and physical properties of the employed indicator, different mechanisms can be responsible for this, which can amplify or counteract each other. [20].

Blood resistivity results predominantly from the electrical resistance of the erythrocytes and is therefore dependent on the hematocrit. Each bolus injection of an erythrocyte-free solution causes hematocrit dilution and consequently a reduced resistivity according to an increased conductivity (conductivity = 1/resistivity) [20]. This effect is enhanced by the use of a hyperosmolar fluid, as intravascular water shift increases hematocrit dilution [20].Blood conductivity is linearly related to ion concentration [20]. Injection of hypertonic ion solution increases blood conductivity, whereas conductivity is decreased by hypotonic or ion-free indicators.

In our experiment, the bolus injection of the two hypertonic sodium-containing solutions (NaCl 5.85% and NaBic 8.4%) led to the optimally evaluable conductivity changes with the highest signal strength and best image quality.

Hypertonic saline has successfully been employed for EIT-based lung perfusion measurement in experimental [10,11,12,13,14] and clinical [15,16,17] studies. Reported concentrations ranged from 3% [12] to 20% [13]. However, the technical advantages of hypertonic saline must be weighed against possible risks and side effects during bolus injection. Administration of hypertonic saline is widely used, e.g., for treatment of increased intracranial pressure [24]. Mostly, rapid infusions of 3% to 7.5% solutions [24], but even much higher doses (intravenous bolus of 30 mL of 23.4% saline [25]), were used in clinical studies and safety was proven [26]. However, experimental data [18] showed that infusion of hypertonic saline (7.5%) decreases vascular resistance and increases myocardial contractility as well as cardiac output. Depending on the extent of these effects, arterial hyper- or hypotension may result. Therefore, the use of hypertonic saline as a contrast agent may affect global and regional hemodynamics and lead to a disturbed measurement situation. Thus, it may influence the results on regional lung perfusion.

The use of hypertonic saline can also lead to chloride overload and hyperchloremia, which is suspected of causing organ dysfunction, such as AKI, due to impaired renal perfusion [19]. However, clinical findings are conflicting. Hyperchloremia has been shown to be common in septic shock and independently associated with AKI. A moderate increase in serum chloride may cause AKI even in patients without hyperchloremia [27]. Accordingly, chloride-restrictive strategies were demonstrated to decrease the incidence of AKI [28]. In contrast, recent analyses of the “HYPR2S” database showed that hyperchloremia is not associated with an increased risk of AKI or mortality despite frequent metabolic acidosis [29]. Finally, Huet et al. [30] recently demonstrated, that hyperchloremia due to increased chloride intake was not associated with AKI in critically ill patients with traumatic brain injury, when treated with hypertonic saline.

In either case, chloride intake can possibly be reduced by using NaBic instead of hypertonic saline. NaBic is widely employed for therapy in critically ill patients with metabolic acidosis [31,32] to normalize pH and pH-depending organ function [31,32]. It is well tolerated during central venous bolus injection [32,33]. However, systematic studies on potential side effects are lacking [32,33]. As NaBic increases PaCO_2_, its use at higher doses could be of concern in hypercapnia, especially if the underlying impairment of pulmonary gas exchange does not allow CO_2_ elimination to be increased by enhanced minute ventilation. Although the evidence for buffering respiratory acidosis with NaBic is inconclusive, this is common in clinical practice [33]. For use as an indicator during EIT-based lung perfusion measurement, the injection of small volumes, as in our experiment, should be unproblematic. Although chloride intake will be reduced, sodium related possible side effects will remain even with the use of NaBic. When choosing sodium containing EIT contrast agents, clinicians should weigh the advantages and disadvantages of each and select according to the patient’s needs.

Finally, a further reduction of the ion load seems generally desirable. This could be achieved by reducing the indicator dose, i.e., by reducing the bolus volume or concentration. Our data (Figure 4b) suggest that a significant reduction in regional signal strength (because of the reduced indicator dose) could be carried out for NaCl- or NaBic-solutions without worsening image quality. Thus, systematical analysis of bolus volume and concentration is needed to best balance reliability and potential side effects.

Glucose 5% is an iso-tonic and non-ionic fluid. After bolus injection, it reduces the ion content of the serum, which decreases blood conductivity. In contrast, hematocrit dilution increases blood conductivity. Since data show an overall decrease in conductivity (Figure 2a) the decrease in plasmatic ion content exceeds the hematocrit effect, which is in line with previous in-vitro experiments [20]. Although regional signal strength was limited, image quality was mostly sufficient. Glucose bolus injections could in principle be used when ion exposure must be avoided. However, success rate, regional signal strength and image quality were poor in a single animal. This may be explained by undiagnosed intracardiac shunt, as the correct timing of the signal sources with contrast-agent flow (right heart, lung, left heart) is a prerequisite for correct functioning of our GD algorithm used for signal extraction [11]. Due to the low regional signal strength after glucose injection, this would result in an inadequate signal-to-noise ratio and inaccurate regional measurements. Regional lung perfusion measurements by glucose bolus injection could therefore be susceptible to interference in disturbed measurement situations. Further studies should investigate whether a modified indicator dosage improves performance.

Jonosteril is an iso-tonic and iso-ionic balanced crystalloid solution. Its injection therefore has minimal effects on the ion concentration of the blood. The influence on blood conductivity is therefore exclusively based on hematocrit dilution as described above. In our experiment, this consequently led to only a small increase in conductivity, the lowest success rates in extracting sufficient data, and the lowest signal strength. In addition, the quality of the images produced was the lowest. In summary, the use of iso-ionic crystalloids, whose contrast signal is based only on hematocrit dilution, seems unsuitable as an indicator for reliable EIT-based measurement of regional lung perfusion.

Bolus injection of Iomeprol caused a clear negative conductivity deflection, which was strong enough to allow reliable extraction of the regional pulmonary perfusion signals. The osmolarity of Iomeprol (≈725 mosmol/kg [34]) is more than twice that of blood (280–300 mosmol/kg). Nevertheless, its conductivity is lower than that of blood because the iodine it contains is in a non-ionized, and thus, non-conducting form. Therefore, the origin of the negative conductivity contrast is the displacement of conducting ions from blood. This displacement effect is enhanced by the high viscosity of Iomeprol (12.6 mPa) [34], which exceeds that of blood (4.5 mPa) by almost three times. Although the image quality was generally good, it was significantly reduced in a single pig. The fact that the same animal was affected in which glucose injection also led to insufficient signals strengthens the suspicion that an anatomical anomaly is responsible (as described above). This suggests that indicators that induce a negative change in conductivity with limited regional signal strength are more susceptible to interference.

Contrast-induced nephropathy may be a severe complication to the administration of iodine-based contrast media for the diagnostic procedure using radiation exposure [35]. Although a network meta-analysis showed that Iomeprol has the best renal safety profile when compared to other iodine-based contrast agent [35], its use outside of contrasted X-ray examinations is prohibitive. However, sequential use of EIT and computed tomography (CT) during an injection of Iomeprol could provide further gains. Combined information on thorax boundary and EIT electrode position can be used to generate a finite-element model (FEM), which increases regional accuracy of EIT reconstruction algorithms [36]. This might further improve the quality of EIT-based perfusion images derived from an Iomeprol bolus. Moreover, CT-based iodine maps could directly be used to validate the results. However, this research might be challenged by different requirements for contrast agent flow profiles during EIT and CT.

Our study clearly has limitations. We conducted these experiments on only five animals. Despite the small number of cases, we were able to obtain enough bolus injections through extensively repeated measurements to make well-founded statements on technical differences of the used indicators regarding success rate, signal strength, and image quality. However, this study was not designed to systematically investigate potential side effects of the respective contrast agents. In addition, we cannot make any statements on the influence of lung damage on the regional measurement results, as the indicators were only examined in healthy lungs. However, the successful use of our gamma decomposition algorithm in measurement situations with disturbed and inhomogeneous lung perfusion has been previously demonstrated [11]. Success rate, signal strength, and image quality were noticeably reduced in a single animal (pig number 3). Although our comments above on the possible anatomical cause (undiagnosed intracardiac shunt) are plausible, they remain speculative and may not fully explain the limited results in this animal. We did not use any other imaging method for validation. Instead, hypertonic saline was used as a reference, for which the suitability has already been demonstrated by our [10] and other [12,13,14] research groups. Our study did not provide a clear alternative that could fully replace hypertonic saline as an EIT contrast agent. Rather, it shows that clinically established solutions can be used in principle and provide acceptable results. Depending on the clinical indication, they could therefore provide an alternative. As with all animal studies, the results should be transferred to patients with caution. Since intrathoracic dimensions, gas and blood volumes differ between pigs and humans, the transfer of the indicator doses and the measured signal strengths is not directly possible. Further studies should research the reliability of these contrast agents in humans. In addition, the influence of changed dosages (injectate volume and concentration) should be systematically investigated.

## 5. Conclusions

In this experimental pilot study, we evaluated the utility of different clinically available injectables as contrast agents for measuring regional lung perfusion using electrical impedance tomography.

The use of isotonic balanced crystalloid solution (Jonosteril^®^) was found to not be helpful, since it technically failed due to a poor success rate of regional perfusion image extraction, low signal strength, and low image quality.

Injections of a non-ionic iodinated X-ray contrast medium (Imeron 400 MCT^®^) and non-ionic glucose solution (Glucose 5%) resulted in mostly well usable signals and sufficient image quality. While Iomeprol might enable simultaneous EIT and X-ray measurements, glucose might help to avoid sodium and chloride overload.

Ionic sodium solutions, such as hypertonic saline (NaCl 5.85%) and sodium-hydrogen carbonate (NaBic 8.4%), led to optimal evaluability with excellent signal strength and image quality. Dosages lower than used in this study might also be applicable.

NaCl 5.8%, NaBic 8.4%, and Glucose 5% seem most suitable for use in the intensive care setting. Further research should address optimal doses to balance reliability and potential side effects.

## Figures and Tables

**Figure 1 jcm-12-02751-f001:**
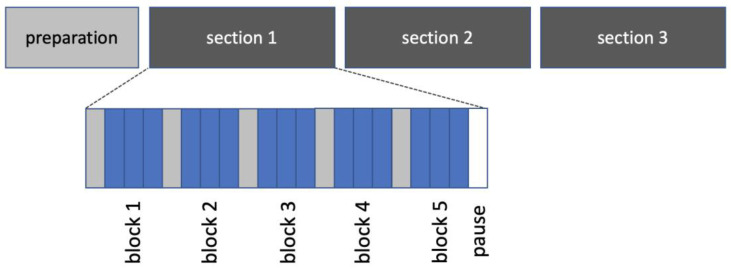
Experimental protocol and randomization. In each animal, measurements were performed in three sections (dark grey). Within any section, the five indicators were tested in blocks of three repeated bolus injections (blue) during temporary apnea in a randomized order. Randomization was performed separately for each section.

**Figure 2 jcm-12-02751-f002:**
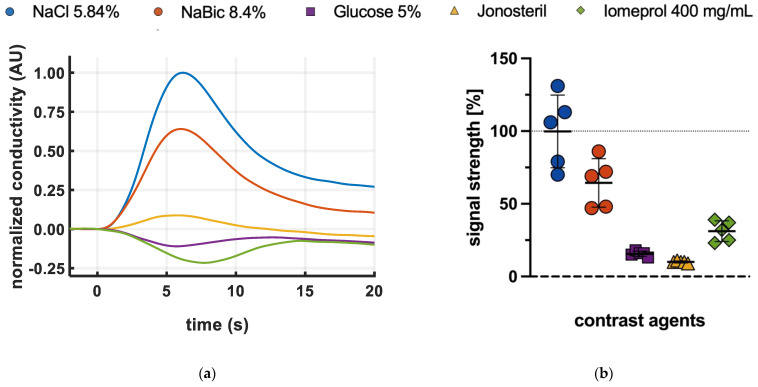
NaCl: sodium chloride, NaBic: sodium bicarbonate (**a**) Exemplary global signals captured by Electrical Impedance Tomography after bolus injection of different contrast agents. Normalized changes in global conductivity over time after filtering and drift compensation; (**b**) relative signal strength of different contrast agents calculated from the maximum amplitude of the regional lung perfusion images. Horizontal solid lines indicate mean values and standard deviation. The dotted line indicates the reference (signal strength of the mean image of NaCl 5.84 injections).

**Figure 3 jcm-12-02751-f003:**
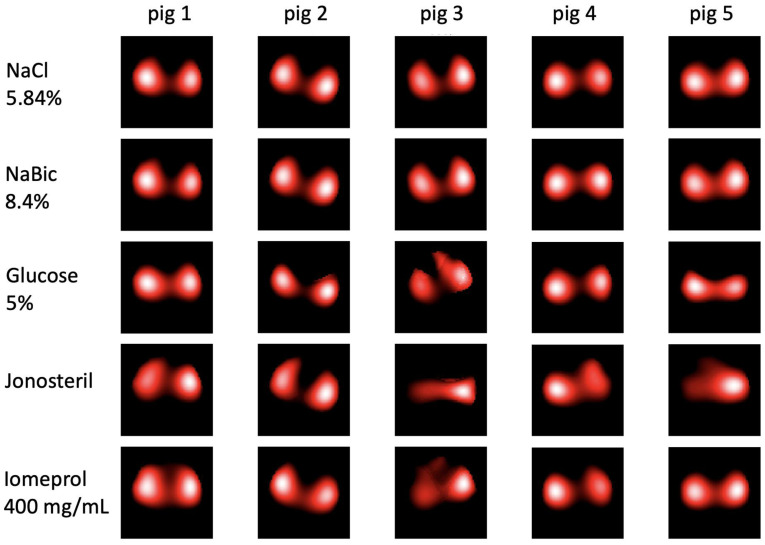
Representative EIT-based lung perfusion images for all animals (1 to 5). Images were reconstructed from bolus injections of the different contrast agents using the Gamma Decomposition algorithm [11]. NaCl: sodium chloride, NaBic: sodium bicarbonate.

**Figure 4 jcm-12-02751-f004:**
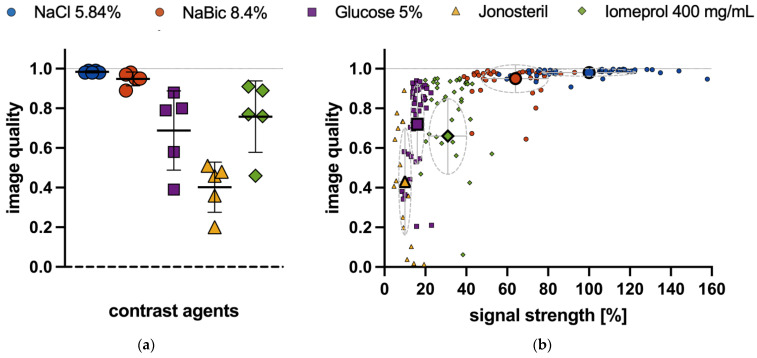
NaCl: sodium chloride, NaBic: sodium bicarbonate (**a**) Image quality estimated by pixel-by-pixel linear correlation analysis (Pearson) calculating image similarity with a reference image. A value of 1 indicates identical images. Each symbol represents one animal. Horizontal solid lines indicate mean values and standard deviation; (**b**) image quality plotted versus signal strength. Small symbols represent one bolus injection. Different contrast agents are grouped. Large symbols represent mean values. Grey crosses and circles indicate corresponding standard deviations in both directions. The dotted line indicates the reference (signal strength of the mean image of NaCl 5.84 injections).

**Table 1 jcm-12-02751-t001:** Success rates of perfusion image extraction obtained for different EIT contrast agents.

Animal	NaCl 5.85%		NaBic 8.4%		Glucose 5%		Jonosteril^®^		Imeron 400 MCT^®^	
	n	%	n	%	n	%	n	%	n	%
1	9	100	9	100	9	100	4	44	8	89
2	9	100	9	100	9	100	6	67	9	100
3	9	100	9	100	4	44	2	22	4	44
4	9	100	9	100	9	100	3	33	9	100
5	9	100	9	100	9	100	4	44	9	100
Mean	9	100	9	100	8	89	3.8	42	7.8	87
SD	0	0	0	0	2.2	25	1.5	16	2.1	24

Data show successful perfusion image extractions in absolute numbers (n) and as proportion (%) of nine repeated injections in each animal. Additionally, mean and standard deviation (SD) are given.

## Data Availability

Algorithms used for EIT image reconstruction are available as part of the EIDORS toolbox [23]. Our model-based source separation algorithm for lung perfusion signals is well described in [11]. The data presented in this study are available in the result section of this article. Raw data are available upon reasonable request from the corresponding author.

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
