# Peer review of "Evaluation of Different Contrast Agents for Regional Lung Perfusion Measurement Using Electrical Impedance Tomography: An Experimental Pilot Study"

_jcm, 2023, doi:10.3390/jcm12082751_

Round 1

Reviewer 1 Report

The Authors conducted an experimental animal study to test the performance of five different agents used as indicators for EIT-based measurement of lung perfusion. Hypertonic saline (NaCl 5.85%) was used as reference. The topic is of interest given the increasing diffusion of EIT and of perfusion assessment. The methods are fine and results quite clear.

Major comments

- while the Authors have to be congratulated for performing many measurements, the clinical relevance of the results is limited because it is difficult to find a clear advantage when using the other tested indicators (maybe with the exception of glucose, which does not perfom too well) instead of hypertonic. Assuming that the small amount used for the bolus does not have relevant side effects, the problems with repeated measurements (and thus with increased cumulative dose of indicators) are not minor with sodium bicarbonate (sodium load + effect on CO2) and Iomeprole (simultaneous EIT perusion and X-ray is not feasible and contrast agents have side effects).

- the poor quality of measurements on animal number 3 remains unexplained

- in the methods I could not find two relevant informations: the body weight of the animals and the duration of the apnea phase

Minor comments:

- Abstract: "Lower dosages of NaCl and sodium-bicarbonate might also provide sufficient and allow repeated measurements" --> this is not a result of the study but an hypothesis and thus it should not be in the abstract

- the supplementary materials (video) need a more detailed explanation (it is not clear what are the raws and the columns, where are the different animals and where is the time..)

Author Response

Reviewer 1

Comments and Suggestions for Authors

The Authors conducted an experimental animal study to test the performance of five different agents used as indicators for EIT-based measurement of lung perfusion. Hypertonic saline (NaCl 5.85%) was used as reference. The topic is of interest given the increasing diffusion of EIT and of perfusion assessment. The methods are fine and results quite clear.

Reply:

We thank the reviewer for her/his nice words and helpful suggestions. Please find our replies below. We hope that all concerns could be addressed.

Major comments

- while the Authors have to be congratulated for performing many measurements, the clinical relevance of the results is limited because it is difficult to find a clear advantage when using the other tested indicators (maybe with the exception of glucose, which does not perfom too well) instead of hypertonic. Assuming that the small amount used for the bolus does not have relevant side effects, the problems with repeated measurements (and thus with increased cumulative dose of indicators) are not minor with sodium bicarbonate (sodium load + effect on CO2) and Iomeprole (simultaneous EIT perusion and X-ray is not feasible and contrast agents have side effects).

Reply:

We agree with the reviewer that our research did not provide a clear alternative that could fully replace hypertonic saline as a contrast agent. We further agree with most of the listed restrictions. However, we would like to make a few clarifications.

Finding an alternative contrast agent as a complete substitute for hypertonic saline was not the aim of our study. Rather, the trials served primarily as a proof of concept for the most promising alternative contrast agents under in-vivo conditions. To the best of our knowledge, there is no such data available in literature so far. However, in order to address the concerns, we have included a comment to this end in the limitation section.

The limitations of the NaBic with regard to the sodium load that persists and the CO2 release have already been explained extensively in the discussion. Nevertheless, the avoidance of chloride load must not be neglected. These advantages and disadvantages must be considered by the clinician when choosing the contrast medium according to the needs of the patient. We have re-emphasised the discussion in this regard.

The side effects of x-ray contrast media have also been discussed. In addition, the technical challenges of the combined use of EIT and CT were discussed. However, we do not agree that simultaneous EIT and CT is not possible. In fact, disturbances in the EIT signal can be triggered by CT. However, both procedures can be performed in direct succession. A quasi-simultaneous use of iomeprol for both procedures is therefore conceivable. For clarification, we have replaced the word "simultaneous" with the word "sequential".

In summary, we have strengthened the discussion on NaBic and Iomeprol and added a general statement in the limitation section.

Changes to the document:

- discussion on NaBic was adapted: “As NaBic increases PaCO2, its use at higher doses could be of concern in hypercapnia, especially if the underlying impairment of pulmonary gas exchange does not allow CO2 elimination to be increased by enhanced minute ventilation. Although the evidence for buffering respiratory acidosis with NaBic is inconclusive, this is common in clinical practice [33]. For use as an indicator during EIT-based lung perfusion measurement, the injection of small volumes, as in our experiment, should be unproblematic. Although chloride intake will be reduced, sodium related possible side effects will remain even with the use of NaBic. When choosing sodium containing EIT contrast agents, clinicians should weigh the advantages and disadvantages of each and select according to the patient's needs. Finally, a further reduction of the ion load seems generally desirable.”

- discussion on Iomeprol was adapted: “However, sequential use of EIT and computed tomography (CT) during an injection of Iomeprol could provide further gains.”

- limitation section was adapted: “Our study did not provide a clear alternative that could fully replace hypertonic saline as an EIT contrast agent. Rather, it shows that clinically established solutions can be used in principle and provide acceptable results. Depending on the clinical indication, they could therefore provide an alternative.”

- the poor quality of measurements on animal number 3 remains unexplained

Reply:

We agree that the results of tier number 3 are suspicious. However, we discuss this issue in detail in the context of the glucose and Iomeprol bolus injection results. We provide a possible explanation, namely that an undiagnosed intracardiac shunt leads to an unexpected sequence of source signals, which affects the gamma decomposition algorithm used. In our opinion, this discussion is appropriate.

However, to again clearly point out the restrictions in this single animal, we have included an additional comment in the limitation section.

Changes to the document:

- limitation section was adapted: “…Success rate, signal strength, and image quality were noticeably reduced in a single animal (pig number 3). Although our comments above on the possible anatomical cause (undiagnosed intracardiac shunt) are plausible, they remain speculative and may not fully explain the limited results in this animal.”.

- in the methods I could not find two relevant informations: the body weight of the animals and the duration of the apnea phase

Reply:

We apologize. Requested information have been added to the manuscript.

Changes to the document:

- section 2.1.:” After approval of the local ethical committee five healthy (2 female, 3 male; 41±5 kg; 13±1 weeks old) pigs were anesthetized and instrumented as previously described [5,21].”

- section 2.2.1.”To access regional lung perfusion, a 10 ml bolus of the respective contrast agent was injected into a central venous line under temporary apnea (60 sec) [11] at PEEP of the preceding ventilation.”

Minor comments:

- Abstract: "Lower dosages of NaCl and sodium-bicarbonate might also provide sufficient and allow repeated measurements" --> this is not a result of the study but an hypothesis and thus it should not be in the abstract

Reply:

We agree. The phrase has been deleted from the abstract.

Changes to the document:

- deleted.

- the supplementary materials (video) need a more detailed explanation (it is not clear what are the raws and the columns, where are the different animals and where is the time..)

Reply:

The reviewer is right. A clear description has been added.

Changes to the document:

- descriptions were added: ”… Each row represents a single animal, pig 1 at top, pig 5 at bottom. In each row, nine repetitive measurements are displayed in chronological order, with the first injection in the leftmost column and the last injection in the rightmost column. All videos are systematically labelled at the top; the labels include the pig number (t1 to t5), the section number (s1 to s3) and the injection number (inj1 to inj3).”

Reviewer 2 Report

Muders et al investigated the suitability of five different injectable and clinically approved solutions as contrast agents for EIT-based lung perfusion measurement with 5 female pigs. They concluded that “while Iomeprol might enable simultaneous EIT and X-ray measurements, glucose might help to avoid sodium and chloride overload. Lower dosages of NaCl and sodium-bicarbonate might also provide sufficient and allow repeated measurements.” There are some concerns that need to be addressed:

Major concerns:

1.      Line 108-111: Please show the repeatability of results using a single contrast agent (e.g. NaCl 5.85%) for the whole sequence in Figure 1 to document that the order of the sequence is not a major factor. Otherwise, the method could be flawed.

2.      Line 155: Data are given as single data points, median and range, or mean and standard deviation. Which data are expressed as median and range, which are mean and standard deviation? How to decide?

3.      Line 157: P < 0.05 was assumed to be statistically significant. Which data were statistically analysed?

4.      Why only female pigs were used? Male pigs should be used as well.

Minor concerns:

5.      Line 71: Spell out the ethical committee number

6.      Line 71: List the ethics approval number

7.      Line 72: Describe the pig strain, age, and body weight

8.      Line 74: Spell out VT and BW

9.      Line 78: Get rid of Appendix A. Instead put the content of Appendix A into the methods of the main text.

10.  Line 82: Section 2.2.1: please describe the repeatability of the method.

11.  Line 93: Spell out in full of NaBic for when it first appears.

12.  English expression could be improved. E.g., “Lower dosages of NaCl and sodium-bicarbonate might also provide sufficient and allow repeated measurements” in the abstract.

Author Response

Reviewer 2

Comments and Suggestions for Authors

Muders et al investigated the suitability of five different injectable and clinically approved solutions as contrast agents for EIT-based lung perfusion measurement with 5 female pigs. They concluded that “while Iomeprol might enable simultaneous EIT and X-ray measurements, glucose might help to avoid sodium and chloride overload. Lower dosages of NaCl and sodium-bicarbonate might also provide sufficient and allow repeated measurements.” There are some concerns that need to be addressed:

 Reply:

We thank the reviewer for her/his helpful suggestions. Please find our replies below. We hope that all concerns could be addressed.

Major concerns:

  1. Line 108-111: Please show the repeatability of results using a single contrast agent (e.g. NaCl 5.85%) for the whole sequence in Figure 1 to document that the order of the sequence is not a major factor. Otherwise, the method could be flawed.

Reply:

We thank the reviewer for this suggestion. We have added some plots to the new Figure S1 of the supplement. In a first step (Figure S1A), we compared results on signal strength of different contrast agents over time using a mixed-effects analysis (mixed models for repeated measurements). The analysis revealed reproducibility over time and showed that differences in signal strength depend on contrast agent but not on time (factor “contrast agent”: p<0.001; factor “time”: p=0.28; interaction “contrast agent*time”: p=0.76). We then plotted individual data points for any single animal and any single contrast agent over time (Figure S1B-F) to visualize repeatability over time.

Changes to the document:

- Figure S1 was added

- statical description was added

- repeatability over time was described in the result section (see below)

  1. Line 155: Data are given as single data points, median and range, or mean and standard deviation. Which data are expressed as median and range, which are mean and standard deviation? How to decide?

Reply:

We present data on signal strength and image quality as scatter plot of animal means. To visualize central tendency of the data, we previously showed the median value. In the scatter plot of signal strength versus image quality we used mean and standard deviation. We agree with the reviewer that, although appropriate according to the different number of data points, mixing up both measures might be confusing. Since we did not perform statistical test on these data, the use of mean and standard deviation is appropriate to visualize central tendency in all plots. We therefore harmonized the plots and show mean and standard deviation only.

Changes to the document:

- the statistic section has been adapted: “Data are given as scatter plots and mean and standard deviation, as indicated.”

- Figure 2b and 4a have been adapted

  1. Line 157: P < 0.05 was assumed to be statistically significant. Which data were statistically analysed?

Reply:

In the previous version this statement described the significance level of the Pearson correlations. In the revised manuscript we added analyses on reproducibility over time. The statement is now valid for both.

Changes to the document:

- none

  1. Why only female pigs were used? Male pigs should be used as well.

Reply:

We regret this embarrassing mistake that occurred during the reworking of the manuscript. In fact, 2 females and three males were studied.

Changes to the document:

- information in section 2.1. was corrected.

Minor concerns:

  1. Line 71: Spell out the ethical committee number

Reply:

According to the JCM guidelines this statement is included at the back matter.

Changes to the document:

- none

  1. Line 71: List the ethics approval number

Reply:

According to the JCM guidelines this statement is included at the back matter.

Changes to the document:

- none

  1. Line 72: Describe the pig strain, age, and body weight

Reply:

In accordance with the reviewer's suggestion, we have deleted Appendix 1 and inserted the information in the methods section. As requested, age and body weight were added.

Changes to the document:

- Information have been added to section 2.1.

  1. Line 74: Spell out VT and BW

 Reply:

We thank the reviewer for this remark and have changed accordingly.

Changes to the document:

- “Tidal volume was 6–8 mL/kg of body weight.”

  1. Line 78: Get rid of Appendix A. Instead put the content of Appendix A into the methods of the main text.

Reply:

In accordance with the reviewer's suggestion, we have deleted Appendix 1 and inserted the information in the methods section.

Changes to the document:

- Information have been added to section 2.1.

  1. Line 82: Section 2.2.1: please describe the repeatability of the method. 

Reply:

As described above, repeatability of measurements is now described in the method and result sections.

  1. Line 93: Spell out in full of NaBic for when it first appears.

Reply:

We agree. The order of the description has been changed.

Changes to the document:

- NaBic (sodium-bicarbonate) 8.4%, ready-to-use hypertonic buffer solution

  1. English expression could be improved. E.g., “Lower dosages of NaCl and sodium-bicarbonate might also provide sufficient and allow repeated measurements” in the abstract.

Reply:

According to the suggestion of Reviewer 1 this phrase has been deleted from the abstract.

Changes to the document:

- deleted

Round 2

Reviewer 2 Report

Thanks for addressing my concerns.